# Parental Knowledge, Attitudes, and Practices and Their Association with Dental Caries in Children Aged 5–9 Years: A Cross-Sectional Study in Rural Ecuador

**DOI:** 10.3390/ijerph22060953

**Published:** 2025-06-17

**Authors:** María Saquicela-Pulla, Mónica Dávila-Arcentales, Eleonor Vélez-León, Ana Armas-Vega, María Melo

**Affiliations:** 1School of Dentistry, Hemisferios University, Quito 170527, Ecuador; anaa@uhemisferios.edu.ec; 2Comprehensive Oral Health Research Group (GIISO), Universidad Politécnica Salesiana, Cuenca 170527, Ecuador; mdavila@ups.edu.ec; 3School of Dentistry, Catholic University of Cuenca, Cuenca 010107, Ecuador; mvelezl@ucacue.edu.ec; 4Department of Stomatology, Faculty of Medicine and Dentistry, University of Valencia, 46010 Valencia, Spain; m.pilar.melo@uv.es

**Keywords:** dental caries, childhood oral health, caregiver practices, oral hygiene, Ecuador, ICDAS, cross-sectional study, health promotion

## Abstract

Childhood dental caries remains a critical public health issue in rural areas, where caregivers’ knowledge, attitudes, and practices (KAP) significantly influence oral health outcomes. This study assessed the association between parental KAP and the prevalence and severity of caries in children aged 5–9 years in Cayambe, Ecuador. Methods: A cross-sectional study was conducted with 229 schoolchildren and their caregivers, using a validated questionnaire and clinical examinations (ICDAS criteria). Caries severity was categorized as “obvious decay” (ICDAS 4–6) or “no obvious decay” (ICDAS 0–3). Results: Advanced dental caries affected 73.4% of children (ICDAS 4–6). While parental knowledge and attitudes showed no significant association, brushing teeth ≥2 times/day significantly reduced caries severity (*p* < 0.05). Sociodemographic factors, such as parental education, were not significant predictors. Conclusion: Oral hygiene practices—particularly frequent toothbrushing—were the key protective factor against severe caries, highlighting the need for behavior-focused interventions promoting daily habits. Future research should evaluate long-term preventive strategies.

## 1. Introduction

Dental caries is a chronic, noncommunicable disease induced by biofilms and modulated by biological, behavioral, psychosocial, and environmental factors [1,2,3,4].

It is especially common in childhood, affecting 514 million children worldwide, according to the World Health Organization (WHO) [5,6,7]. Its negative impact is reflected in child development, affecting quality of life, well-being, nutrition, growth, speech, school performance, and self-esteem [6]. Therefore, dental caries is considered a public health problem [8]. Without treatment, it can lead to pulp lesions and premature tooth loss [9,10].

Poor oral hygiene practices in children are associated with a higher prevalence and severity of caries [11,12], highlighting the importance of preventive strategies focused on educating parents, who are crucial in transmitting healthy habits [13,14,15,16]. During the school years, children develop autonomy, language, and habits [17], making oral care a priority for professionals. However, mixed dentition and dental changes pose a challenge [18,19].

An integral approach to preventing and treating caries involves active parental participation through educational strategies and habit promotion [20,21]. Studies in Ecuador emphasize the importance of instilling these habits from early stages, even before tooth eruption, due to their significant impact on the quality of life of children and their families [22]. Implementing multidisciplinary efforts with periodic monitoring is recommended to manage the disease [23] effectively.

The family environment is crucial for children’s oral health [24]. Parental education, socioeconomic level, and prevention awareness are associated with higher caries rates [25,26]. Studies in Ecuadorian indigenous populations have highlighted that low maternal education is related to poor oral health in children [27]. Research in China [28] and India [29] also shows that the educational level of parents and parenting style significantly impact children’s oral health, while in Australia [30] and Italy [31], it is observed that children of parents with lower educational levels are more prone to dental problems due to a lack of knowledge and proper practices. Parental oral health literacy, defined as the ability to understand and apply relevant information in dental health decisions, is closely linked to the incidence of childhood caries [32].

Evidence suggests a “triple disadvantage” in vulnerable contexts: low educational level, belonging to migrant communities, and living in impoverished neighborhoods, factors that exacerbate the risk of caries [33]. Misconceptions about primary dentition also contribute to carious lesions [26].

The ICDAS index provides a standardized methodology for value-dating oral health, enabling more effective public health interventions [34]. In Ecuador, childhood caries remains an alarming public health issue. A 2023 study revealed that 75.2% of preschoolers in three Ecuadorian communities had caries [23], while in the Galápagos Islands, 68% of the schoolchildren examined had active carious lesions, with more than 30% requiring urgent treatment [35]. These figures underscore the need for preventive interventions focused on the role of parents.

The present study aims to determine the relationship between the knowledge, attitudes, and practices (KAP) of parents of children aged 5 to 9 at the Remigio Crespo Toral school and the prevalence and severity of dental caries to develop more targeted and effective prevention strategies. The primary objective is to assess this association in a rural Ecuadorian population, while the secondary objective is to identify which specific KAP components most predict advanced carious lesions. The null hypothesis (H_0_) suggests no statistically significant association between caregiver KAP variables and the severity of dental caries, as measured by the ICDAS-II system. This study also aims to inform the design of evidence-based educational strategies in similar vulnerable populations.

## 2. Materials and Methods

### 2.1. Study Design and Ethical Considerations

A cross-sectional observational study was conducted at the Remigio Crespo Toral Educational Unit in Cayambe, Pichincha, Ecuador. The target population was schoolchildren aged 5 to 9 and their primary caregivers. The UNIANDES University Human Research Ethics Committee (CEISH-UNIANDES, approval code: 2024-EXT-OB-0015) granted ethical approval. Written informed consent was obtained from all caregivers before participation in compliance with the Declaration of Helsinki.

### 2.2. Study Variables

The primary variable was the presence of dental caries, which was clinically evaluated using the ICDAS-II system. For the analysis, the findings were grouped into two categories:No obvious decay (codes 0 to 3): Includes healthy teeth or teeth with initial lesions without cavitation.Obvious decay (codes 4 to 6): Corresponds to cavitated lesions involving dentin, with direct clinical involvement.

The explanatory variables were obtained through a structured questionnaire, self-administered by the caregivers, which explored sociodemographic aspects, knowledge, attitudes, and practices in children’s oral health (CAP model).

Sociodemographic data: Included the child’s age and sex.

Knowledge:

Fourteen statements about children’s oral health (fluoride use, bacterial transmission, nighttime breastfeeding, cariogenic foods, etc.) were evaluated. Correct answers scored one point each, and knowledge was classified as adequate if at least 11 correct answers were obtained.

b.Attitude:

Caregivers indicated how important they considered the child’s oral health. For the analysis, a distinction was made between favorable Attitude (“very important”) and other responses.

c.Practices:

They included habits such as the child’s feelings when brushing (Happy, Neutral, Resistant, Sad) and whether the child brushes his or her tongue (Yes, No, Do not know).

These variables about caries severity were analyzed individually and in multivariate models to identify the most relevant factors in protecting against the disease.

### 2.3. Inclusion and Exclusion Criteria

Inclusion criteria

Children aged 5–9 years enrolled in school.Informed consent signed by caregivers.

Exclusion criteria

▪Children with severe systemic diseases (e.g., congenital heart disease, immune disorders, uncontrolled diabetes, or ongoing cancer treatment).▪Children undergoing specialized or ongoing dental treatment (orthodontics, endodontics, or post-surgical care).▪Absence of informed consent.

### 2.4. Population and Sample

The target population of the present study was composed of school children between 5 and 9 years of age, residents of Cayambe, located in the northern region of Ecuador. To determine the required sample size, an estimated prevalence of dental caries of 75% was assumed based on national epidemiological data previously reported in the Ecuadorian child population [23]. A confidence level of 95% and a margin of error of 5% was established.

Considering the urban study area’s finite population of approximately 560 schoolchildren, the corresponding adjustment was applied, obtaining a minimum necessary size of 207 participants.

An additional 10% margin was incorporated to ensure statistical validity in the face of possible loss of information or non-response. Thus, the final sample size was established at 229 schoolchildren, a number that was reached in full during the fieldwork.

This calculation ensured a statistical power of more than 80% to identify associations of moderate magnitude in bivariate analyses (such as the chi-square test) and logistic regression models with dichotomous variables under the assumption of a balanced distribution of exposures and sufficient variability of the primary outcome.

### 2.5. Calibration of Examiners and Data Collection Protocol

To guarantee the quality, consistency, and reliability of the study’s clinical and social data collection, a training and calibration process was implemented for the research team, which consisted of five examiners with dental training.

The team was trained in using the ICDAS-II system for the clinical detection of caries lesions through theoretical and practical sessions led by the principal investigator. The theoretical phase included a detailed review of the diagnostic criteria, complemented by the analysis of standardized clinical images and reference cases. Subsequently, a practical phase was carried out with 20 simulated cases that included different degrees of severity in primary and permanent dentition, replicating field conditions.

Diagnostic agreement between examiners was evaluated using Cohen’s kappa coefficient, obtaining an overall reliability of κ = 0.87, representing an almost perfect agreement level when the results were disaggregated, κ = 0.83 for the primary dentition and κ = 0.86 for the permanent dentition, values that support the consistency of the clinical diagnosis among the evaluators.

At the same time, the team was also instructed to apply, the KAP (knowledge, attitudes, and practices) questionnaire, an instrument designed for the evaluators. This questionnaire was culturally adapted from previous validated models [21] and reviewed by public health and pediatric dentistry experts. It included sections on sociodemographic data, knowledge of children’s oral health, perception of importance (attitudes), and oral hygiene practices at home. It was self-administered in a supervised setting, with neutral guidance from trained personnel to avoid bias or direct interventions.

Before the fieldwork, a pilot test was conducted with 20 caregivers outside the final sample. This made it possible to verify the questionnaire’s understanding, adjust the wording, and improve its adaptation to the local context. This validation process strengthened the quality of the data obtained and the team’s application experience.

### 2.6. Data Collection Procedures

Data were collected during the second semester of 2024 in Cayambe, a city in northern Ecuador. The process was carefully planned to guarantee adequate conditions for both the clinical evaluations and the questionnaire application, respecting the dynamics of the households and the local educational environment at all times.

All the necessary institutional authorizations were obtained before starting the fieldwork. Likewise, caregivers signed an informed consent form, thus ensuring their voluntary participation and compliance with international ethical research principles, including confidentiality and anonymity.

The collection was organized in two complementary stages. First, the clinical evaluation of the children was carried out in a space adapted to provide privacy and biosafety conditions. Five previously trained and calibrated examiners evaluated each participant using the ICDAS-II system. Dental surfaces were carefully inspected using basic instruments (mirror, probe, headlight) and recorded according to standardized diagnostic criteria. Subsequently, the findings were grouped into two categories: No obvious decay (ICDAS 0–3) and obvious decay (ICDAS 4–6).

In parallel, the caregivers autonomously answered the CAP questionnaire, designed to explore their knowledge, attitudes, and practices related to their children’s oral health. The sex application was conducted in a quiet environment, with the research team accompanying to resolve doubts without intervening in the answers. The filling-out time was short, and the answers were reviewed promptly to ensure completeness.

At the end of the process, the data were digitized with a double check and stored in an anonymized database with restricted access. This approach ensured careful, respectful, and methodologically sound data collection that was aligned with the study’s objectives and the particularities of the local context.

### 2.7. Data Management and Analysis

Once data collection was completed, a systematic review, coding, and digitalization were carried out. The research team carefully checked all clinical and KAP questionnaire forms to identify inconsistencies, omissions, or recording errors. Subsequently, the data were entered into a structured database using Microsoft Excel (version 2505), with independent double entry and subsequent reconciliation to ensure accuracy.

The final database was anonymized, numerical codes assigned to each participant and password-protected on institutional devices for the exclusive use of the research team. This strategy maintained the information’s confidentiality and guaranteed the records’ integrity.

The statistical analysis was carried out using IBM SPSS Statistics v.27 software. First, a descriptive analysis of the sociodemographic, clinical, and behavioral variables was performed. Categorical variables were expressed as absolute frequencies and percentages; continuous variables were expressed as central tendency and dispersion measures according to their distribution.

To evaluate associations between the presence of caries (dichotomous dependent variable) and the explanatory variables (knowledge, attitudes, and practices), bivariate Chi-square tests (χ^2^) were used. A binary logistic regression model included the variables that presented a statistically significant association (*p* < 0.05) and those clinically relevant according to the literature. Odds ratios (ORs) were estimated with their respective 95% confidence intervals, making it possible to identify factors associated with a greater or lesser probability of presenting advanced caries.

Statistical significance was established at a 95% confidence level. Cautious interpretation criteria were applied throughout the analysis, considering the study’s cross-sectional nature and the possibility of non-causal relationships.

## 3. Results

### 3.1. Sample Distribution by Age and Sex

Regarding sex, 52.4% of the participants were male (N = 120), while 47.6% were female (N = 109) with a total sample size of N = 229. This demographic breakdown ensures representative coverage across early childhood school ages and sexes, supporting the reliability of comparisons and further subgroup analysis in the study. Table 1 shows the sample distribution by age and sex.

### 3.2. Classification of Dental Caries Severity According to ICDAS Grouping

Table 2 shows that of the 229 children clinically evaluated, 73.4% (N = 168) showed lesions with cavitation or structural deterioration, classified as “obvious decay” (ICDAS 4–6). Caries severity was analyzed using non-parametric tests (Chi-square and Fisher’s exact test) and logistic regression. In contrast, 26.6% (N = 61) were classified as having no obvious decay, which includes those without visible lesions (ICDAS 0) or with initial signs of enamel demineralization without loss of structural integrity (ICDAS 1–3).

### 3.3. KAP Survey Instrument (Knowledge, Attitudes, and Practices)

Table 3 summarizes the responses from a KAP questionnaire administered to parents or caregivers regarding children’s oral health. The responses are grouped by the KAP domain (knowledge, attitudes, or practices) and the associated variable. Each row displays the response category, and the percentage within the specific variable.

### 3.4. Bubble Plot of Logistic Regression Coefficients for Caregivers’ Knowledge Items Associated with Obvious Decay in Children (ICDAS 4–6)

Figure 1 presents a bubble plot illustrating the association between caregivers’ responses to 14 knowledge items related to children’s oral health and the presence of obvious dental caries in their children (ICDAS codes 4–6). The vertical axis displays the 14 knowledge items, while the horizontal axis reflects the corresponding response categories. Each bubble represents a regression coefficient (β), with color indicating the direction of association (blue for negative/protective, red for positive/risk-associated) and bubble size proportional to statistical significance, with larger bubbles representing lower *p*-values (*p* < 0.10).

The most visually impactful associations include the following: The response “Between meals” as the best time to consume sweets, which appears as a prominent blue bubble (β = –0.92, *p* = 0.07), indicating a significant negative association with the presence of caries—interpreted as a potentially protective behavior. Interestingly, the response “between meals” as the best time to consume sweets appeared to have a statistically significant negative association with caries presence, which is counterintuitive. This may be due to confusion in interpreting the question or a reflection of complex dietary patterns not captured in detail by the survey instrument. Therefore, this finding should be interpreted with caution and may merit further investigation.

The lack of knowledge regarding cleaning baby teeth with gauze (“I do not know”), which is positively associated with caries presence (β = 0.62, *p* = 0.08), highlights a critical gap in preventive knowledge. Additional noteworthy items include a lack of awareness regarding appropriate actions during dental pain and uncertainty about the effects of nighttime bottle use, represented by medium-sized red bubbles—suggesting increased risk due to insufficient knowledge in these domains

Conversely, responses aligned with evidence-based recommendations—such as fluoride toothpaste, routine toothbrush replacement, and brushing three times daily—are represented by blue bubbles, though smaller in size, indicating protective trends that did not reach statistical significance in this analysis. This visualization effectively identifies the direction and strength of associations, helping pinpoint knowledge gaps that may be most relevant for targeted caregiver education aimed at preventing early childhood caries.

### 3.5. Association Between Caregivers’ Attitude Toward Oral Health and the Presence of Obvious Decay

Table 4 shows that among the 229 caregivers surveyed, 73.8% (N = 169) reported considering their child’s oral health as “very important.” In contrast, 21.0% (N = 48) classified it as “important,” 4.8% (N = 11) as “neutral,” and 0.4% (N = 1) as “less important.”

To evaluate whether this self-reported attitude was associated with the presence of obvious decay in children (defined as at least one occlusal surface with an ICDAS code 4–6), a bivariate logistic regression analysis was conducted comparing those who rated oral health as “very important” versus all other categories combined.

The model yielded an odds ratio (OR) of 1.32 with a 95% confidence interval (CI) of 0.73–2.37 and a *p*-value of 0.363. This result indicates no statistically significant association between a highly positive caregiver attitude and a lower likelihood of caries presence. Although the direction of the association was positive, the wide confidence interval crossing the null value suggests that the observed difference may be due to chance. These findings imply that, while most caregivers express a strong appreciation for oral health, this attitude alone does not appear to translate into a lower prevalence of advanced carious lesions in children—underscoring the importance of promoting awareness and consistent implementation of preventive oral health practices.

### 3.6. Frequency and Association of Children’s Oral Health Practices (N = 229)

Table 5 shows that most caregivers reported that their children experience positive emotions during tooth brushing, with 91.3% of children described as “happy” while brushing. Only 7.9% showed resistance, and a minimal percentage (0.9%) felt sad during this activity. Regarding tongue hygiene, 67.2% of children brushed their tongues, 19.2% of caregivers were unsure, and 13.5% indicated their children did not.

When analyzing the association between these practices and the type of caregiver, being a mother was not significantly related to a higher likelihood of the child demonstrating positive oral hygiene practices, such as experiencing a pleasant emotional state during brushing (OR = 0.30; 95% CI: 0.07–1.35; *p* = 0.117). Similarly, tongue brushing did not significantly correlate with the mother being the primary caregiver (OR = 0.81; 95% CI: 0.42–1.54; *p* = 0.519).

### 3.7. Forest Plot—Multivariate Model (KAPs → Advance)

Figure 2. When jointly evaluating the components of the KAP model, it was found that only the proper brushing practice (≥2 times a day) was significantly associated with a lower likelihood of having obvious cavities, with an adjusted OR of 0.29 (95% CI: 0.13–0.64; *p* = 0.002). This indicates that children with adequate brushing frequency had a 71% lower risk of developing advanced lesions than those who brushed once a day or less.

On the other hand, both adequate knowledge and a highly positive attitude toward oral health showed non-significant protective associations (*p* = 0.090 and *p* = 0.072, respectively). These findings suggest that while knowledge and attitude may influence behavior, concrete practice serves as the determining factor in the prevention of cavities in this sample.

## 4. Discussion

The results of the study show that, although the null hypothesis (suggesting that there was no association between parental knowledge, attitudes and practices and childhood caries) is partially fulfilled by not finding a significant relationship between knowledge/attitudes and caries reduction, it is rejected in the specific case of hygiene practices. Frequent tooth brushing (≥2 times/day) proved to be a significant protective factor, reducing the risk of advanced caries by 71% (OR = 0.29, *p* = 0.002), which is evidence that, beyond theoretical knowledge or favorable attitudes, it is the concrete actions of oral care that impact caries prevention in children.

### 4.1. Comparison of Results

The findings of this study confirm a widely documented reality in the region: dental caries in school-age children continues to be a chronic and prevalent problem, particularly in rural areas. More than 70% of the children in this Ecuadorian sample presented obvious decay caries (ICDAS 4–6). This result resembles reports in other communities in the country and Latin America, where the figures exceed 75% in preschool and school-age children [20,22,35]. Similarly, studies in Brazil, Colombia, Mexico, and even more distant regions such as the Galapagos Islands have documented prevalences close to or above 80% [11,26,36]. These data show that beyond the generalized knowledge of the importance of oral hygiene, structural, economic, and behavioral barriers continue to limit its application.

A particularly revealing finding was that brushing teeth at least twice daily was significantly associated with a lower prevalence of advanced caries. At the same time, the caregivers’ knowledge levels and positive attitudes did not show a statistically significant relationship. This coincides with studies carried out in Mexico, South Africa, and the Middle East, where it has been shown that theoretical knowledge alone does not guarantee effective practices [21,37,38,39]. This phenomenon, known as the “know-do gap,” has been reported in various public health contexts. In other words, caregivers may know that sugar causes caries or that brushing is important, but this does not necessarily imply that they effectively supervise their children’s brushing or restrict the consumption of sweets [32].

Specific differences in practical knowledge were also identified. For example, lack of knowledge about cleaning with gauze in infants or what to do when faced with dental pain in children was associated with a greater presence of caries, although without reaching statistical significance. These results suggest that specific knowledge gaps could be addressed through targeted interventions, particularly during the first years of life [14,40,41].

On the other hand, the educational level of the caregivers, a variable that in many studies is associated with better results in children’s oral health [21,34], did not show a significant association in this case. One possible explanation lies in the socio-educational homogeneity of the sample, which could have limited the detection of differences. In addition, the impact of educational level may operate indirectly, facilitating more consistent practices but not necessarily guaranteeing positive results if these practices are not implemented.

Additionally, the analysis of sociodemographic variables showed that the caregivers’ level of education was not significantly associated with caries prevalence in children (OR = 1.08, 95% CI: 0.71–1.65, *p* = 0.723), indicating that other behavioral or contextual factors may play a more decisive role in this population.

### 4.2. Instruments and Methodology

This study used two fundamental tools: a culturally adapted and expert-validated KAP (knowledge, attitudes, and practices) survey and a clinical caries assessment using the ICDAS-II system. KAP surveys have proven useful in characterizing family dental public health settings, and their validity in similar contexts has been confirmed in research conducted in South Africa and Iran [21,34]. However, as with any self-reported instrument, the potential for social desirability or recall bias is recognized, particularly in the practice sections.

Diagnosis with ICDAS-II was a methodological strength. Unlike the DMFT index (which stands for decayed, missing, and filled teeth and is widely used in epidemiological studies), this system allows for detecting caries in the early stages and classifying severity more accurately [20]. This approach has recently been adopted in studies in Ecuador and other regions of Latin America [8,20,36], demonstrating that its application is feasible even in rural contexts if adequate training is available, as was done in this study with an index κ > 0.80.

### 4.3. Limitations and Biases

Among this study’s limitations is its cross-sectional design, which prevents us from establishing causal relationships. In addition, since the sample was obtained from a single educational unit, the results are not generalizable to other populations with different socioeconomic or cultural conditions. The possible selection bias is also relevant since caregivers more interested in their children’s health could have been more willing to participate.

Limitations are also recognized in measuring variables such as diet or access to fluoride sources, which were not quantified in detail and could act as confounding factors. Finally, the caries observed could result from previous habits, not current ones, an inherent limitation of cross-sectional studies in child populations.

### 4.4. Study Contributions

Despite its limitations, this study provides relevant evidence for the Ecuadorian context, showing that oral hygiene practices, rather than knowledge or attitude alone, directly impact children’s oral health. It also contributes to the debate on the effectiveness of traditional educational strategies, suggesting the need for programs that inform, facilitate, and encourage concrete behaviors in the home.

Applying ICDAS-II and validating a KAP instrument in the Ecuadorian rural context also represents a methodological contribution to future research, especially in Latin America, where standardized tools are limited.

Finally, this study strengthens the call to design culturally adapted interventions that promote sustainable changes in family environments. Childhood caries figures remain alarming in Ecuador and the region, but this study provides a deeper understanding of the critical points where health promotion can act.

## 5. Conclusions

The study highlights the critical prevalence of childhood dental caries in rural areas of Ecuador and underlines that, although caregivers’ knowledge and attitudes towards oral health do not significantly correlate with the presence of caries, oral hygiene practices, especially frequent brushing (more than twice a day), are key protective factors. These findings emphasize the need for behaviorally focused interventions that encourage daily oral hygiene habits, suggesting that future research should focus on long-term preventive strategies that encourage effective practices from an early age.

## Figures and Tables

**Figure 1 ijerph-22-00953-f001:**
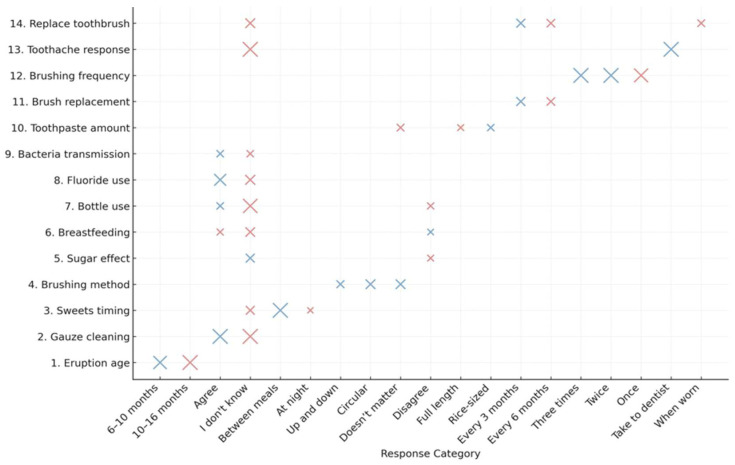
Bubble plot of logistic regression coefficients for caregivers’ knowledge items associated with obvious decay in children (ICDAS 4–6). Note: Bubble plot representing β coefficients from logistic regression models examining the association between caregivers’ knowledge responses and the presence of obvious decay (ICDAS codes 4–6) in children. Each bubble represents a unique response category. The vertical axis shows the survey item number and theme, and the horizontal axis shows the response options. The color indicates the direction of association (blue = negative, red = positive), and the bubble size reflects the statistical significance (larger size = lower *p*-value).

**Figure 2 ijerph-22-00953-f002:**
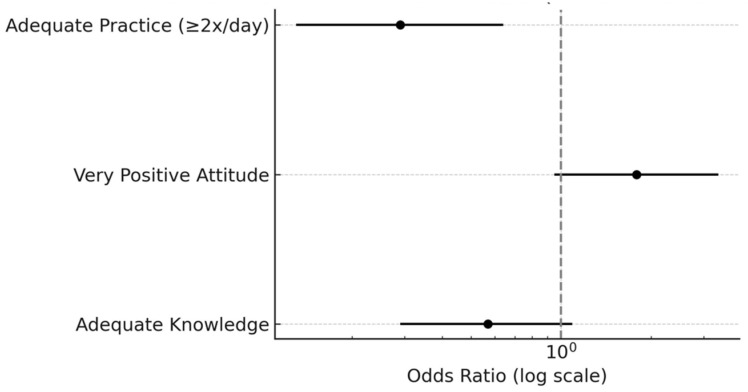
Forest plot—multivariate model (KAPs → advance). Notes: The results of this study reject the null hypothesis, demonstrating a statistically significant association between caregivers’ oral hygiene practices—specifically, the frequency of tooth brushing—and the presence of obvious decay ICDAS 4–6 in children. In contrast, no significant associations were found between the severity of cavities and caregivers’ knowledge or attitudes. These findings highlight the predominant role of behavioral practices in determining oral health outcomes in early childhood, even when general knowledge and attitudes toward oral health are favorable.

**Table 1 ijerph-22-00953-t001:** Sample distribution by age and sex.

Age (Years)	Male N (%)	Female N (%)	Total N (%)
5	3 (1.3%)	3 (1.3%)	6 (2.6%)
6	14 (6.1%)	9 (3.9%)	23 (10.0%)
7	38 (16.6%)	43 (18.8%)	81 (35.4%)
8	37 (16.2%)	26 (11.4%)	63 (27.5%)
9	28 (12.2%)	28 (12.2%)	56 (24.5%)
Total	120 (52.4%)	109 (47.6%)	229 (100%)

Abbreviations: N: number of children, %: percentage calculated over the total sample (N = 229). The category “9” includes all children aged 9 years or older. Percentages may not total exactly 100% due to rounding.

**Table 2 ijerph-22-00953-t002:** Classification of dental caries severity according to ICDAS grouping.

ICDAS Code Range	Decay Classification	N	%
0–3	No obvious decay	61	26.6
4–6	Obvious decay	168	73.4
	Total	229	100

Note: ICDAS: International Caries Detection and Assessment System. N: number of children. %: percentage. Obvious decay: presence of at least one lesion classified as ICDAS codes 4–6. No obvious decay: includes children with ICDAS codes 0–3 (sound teeth or non-cavitated enamel lesions).

**Table 3 ijerph-22-00953-t003:** KAP survey instrument (knowledge, attitudes, and practices).

KAP Category	Variable	Response Category	%
**Attitude**	Attitude toward dental health	Important	21.0
		Less important	0.4
		Very important	73.8
		Neutral	4.8
**Practice**	Child’s feeling when brushing teeth	Happy	90.8
		Neutral	0.4
		Resistant	7.9
		Sad	0.9
	Child brushes the tongue	No	13.5
		I don’t know	19.2
		Yes	67.2
**Knowledge**	Foods that cause cavities	Chocolate	52.0
		Fruits	4.8
		Cookies	29.3
		Cakes	14.0
	Prolonged breastfeeding causes cavities	Agree	26.6
		Disagree	21.4
		I don’t know	52.0
	Amount of toothpaste used	Full length of brush	65.5
		It doesn’t matter	3.5
		Pea size	31.0
	Excess sugar causes cavities	Agree	85.6
		Disagree	2.2
		I don’t know	12.2
	Sleeping with bottle causes cavities	Agree	45.0
		Disagree	5.7
		I don’t know	49.3
	Age of first tooth eruption	10–16 months	18.8
		6–10 months	71.6
		I don’t know	9.6
	Cleaning baby teeth with gauze	Agree	68.1
		Disagree	6.6
		I don’t know	25.3
	Best time to give sweets/cold drinks	After meals	53.3
		Between meals	12.7
		None	0.4
		I don’t know	33.2
		Evening	0.4
	Brushing method	Forward/backward	16.6
		Up/down	39.7
		Circulars	37.6
		It doesn’t matter	6.1
	Transmission of bacteria from mother to child	Agree	58.1
		Disagree	9.6
		I don’t know	32.3
	Use of fluoride toothpaste	Agree	54.1
		Disagree	5.2
		I don’t know	40.6
	Toothbrush replacement frequency	Every 3 months	77.3
		Every 6 months	16.6
		When they open	3.9
		I don’t know	2.2
	Brushing frequency	After every meal	17.9
		Twice	16.2
		Thrice	63.3
		Once	2.6
	What to do if child has toothache	Analgesic	10.0
		Salt water	9.6
		Take to the dentist	76.4
		I don’t know	1.7
		Eat elsewhere	2.2

Percentages (%) are calculated within each variable. KAP = knowledge, attitudes, and practices.

**Table 4 ijerph-22-00953-t004:** Caregivers’ attitude toward oral health and its association with obvious decay (ICDAS 4–6).

Attitude Toward Oral Health	Frequency	Percentage (%)	OR	*p*-Value
Very Important	169	73.8	1.32	0.363
Important	48	21.0		
Neutral	11	4.8		
Less Important	1	0.4		

Notes: Frequencies and percentages were derived from a total of 229 caregivers. The odds ratio (OR) and *p*-value were calculated using logistic regression comparing caregivers who considered oral health as “Very Important” versus all other response categories combined. Obvious decay was defined as ICDAS codes 4–6. No statistically significant association was observed (*p* = 0.363).

**Table 5 ijerph-22-00953-t005:** Frequency and association of children’s oral health practices (N = 229).

Variable	Response	Frequency	Percentage (%)	OR (95% CI)	*p*-Value
Child’s feeling when brushing teeth	Happy	209	91.3%	0.30 (0.07–1.35)	0.117
	Neutral	0	0.0%		
	Resistant	18	7.9%		
	Sad	2	0.9%		
Child brushes the tongue	Yes	154	67.2%	0.81 (0.42–1.54)	0.519
	I don’t know	44	19.2%		
	No	31	13.5%		

Notes: Frequencies reflect caregiver-reported oral health practices. OR = odds ratio from logistic regression using caregiver type (mother vs. other) as a predictor. CI = confidence interval. Statistical significance was set at *p* < 0.05.

## Data Availability

The data presented in this study are available upon request from the corresponding author. While the data have been anonymized, the ethics committee has not authorized public release at this time. The dataset is currently reserved for editorial review purposes.

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
