# Peer review of "Parental Knowledge, Attitudes, and Practices and Their Association with Dental Caries in Children Aged 5–9 Years: A Cross-Sectional Study in Rural Ecuador"

_ijerph, 2025, doi:10.3390/ijerph22060953_

Round 1
Reviewer 1 Report
Comments and Suggestions for Authors
I read the manuscript titled "How Parental Knowledge, Attitudes, and Practices Shape Childhood Dental Caries: Evidence from a Rural Ecuadorian Population Using the ICDAS Index" with great interest and curiosity. This study is of particular significance as it investigates a rural population in South America—a region often underrepresented in the literature.
After thoroughly reviewing the manuscript and the study itself, I have identified several critical issues that currently prevent the manuscript from being suitable for publication. However, I believe that, with comprehensive and appropriate revisions, the likelihood of the manuscript becoming publishable could significantly increase. In this context, I would like to outline my revision requests as follows:
-
Beginning the title with “How” detracts from the scientific rigor of the study. I recommend adopting a clearer title that reflects the study design (e.g., cross-sectional study) and omitting the term “ICDAS” from the title.
-
The Materials and Methods and Results sections of the Abstract need further elaboration. Methodological details and key findings should be stated more explicitly.
-
Citations from recent literature are always more valuable. In this regard, I strongly suggest incorporating the following article, authored by a group of editors, in the first part of the Introduction, especially for context on dental caries and erosion:
Campus G, Niu JY, Sezer B, Yu OY. Prevention and management of dental erosion and decay. BMC Oral Health. 2024 Apr 17;24(1):468. doi: 10.1186/s12903-024-04257-y. PMID: 38632545.
-
While it is commendable that you used a contemporary definition of dental caries, I also recommend referencing the following articles for a more robust and updated framework:
Monteiro ELO, Ladeira LLC, Costa CM, Monteiro MCC, Rebelo MAB, Alves CMC, Thomaz EBAF, Ribeiro CCC. Behavior Risk Factors for Noncommunicable Diseases and Caries in Adolescents: A Population Study. Caries Res. 2025 Feb 18:1-10. doi: 10.1159/000544723;
Pitts NB, Zero DT, Marsh PD, Ekstrand K, Weintraub JA, Ramos-Gomez F, Tagami J, Twetman S, Tsakos G, Ismail A. Dental caries. Nat Rev Dis Primers. 2017 May 25;3:17030. doi: 10.1038/nrdp.2017.30;
Pitts NB, Twetman S, Fisher J, Marsh PD. Understanding dental caries as a non-communicable disease. Br Dent J. 2021 Dec;231(12):749-753. doi: 10.1038/s41415-021-3775-4.
-
I noticed that several of your citations are in local languages (e.g., Spanish or Portuguese). While these are certainly valuable, please keep in mind that you are submitting to a high-impact international journal. Therefore, more references should be drawn from peer-reviewed, English-language journals with broader international recognition.
-
Clarify whether the index you used was ICDAS or ICDAS-II. Since ICDAS has been revised as ICDAS-II, it is essential to specify and describe the version used.
-
In the Introduction, please elaborate on the consequences and implications of untreated dental caries. Moreover, provide further information on the risk factors contributing to caries development and current preventive strategies.
-
Clearly state the null hypothesis, as well as the primary and secondary outcomes, at the end of the Introduction. Additionally, the first paragraph of the Discussion section should clearly state whether the null hypothesis was accepted or rejected.
-
The caries index used must be described in much greater detail.
-
The Sample Size Calculation section is insufficient. Please include the referenced values (e.g., mean, standard deviation, effect size) and describe the rationale behind your calculations in more detail.
-
The Parental Oral Health Knowledge section also needs substantial expansion. What were the contents of the questions? How was the scoring done? How was the total score calculated? How were the questions developed—was a pilot study conducted? Who conducted the interviews and examinations? Were inter-rater agreement values calculated?
-
Overall, the Methods section is inadequate and must be expanded to include all relevant details.
-
Some figure captions are written in a local language. Please ensure that the entire manuscript is professionally edited for grammar, punctuation, and overall English language usage.
-
The Discussion section is currently very brief and superficial. There are hundreds—if not thousands—of studies examining the prevalence of dental caries and its contributing factors in children. Please review recent literature thoroughly and update this section accordingly, placing your findings in a broader research context.
-
As it currently stands, the manuscript is unfortunately not suitable for publication. I believe that all of the above revision requests must be addressed comprehensively. I would be happy to re-evaluate the manuscript after these revisions are completed.
Wishing you continued success in your research.
Comments on the Quality of English LanguageSome sections of the manuscript contain text written in a local language (likely Portuguese or Spanish). Additionally, a thorough check of the English grammar and punctuation is essential.
Author Response
Title of the Study: Thank you for your observations regarding the title. We have changed the title to "Parental Knowledge, Attitudes, and Practices and Their Association with Dental Caries in Children Aged 5–9 Years: A Cross-Sectional Study in Rural Ecuador," which clearly reflects the study design.
Materials and Methods and Abstract Results: We appreciate your suggestion and have further developed and clarified the methodological details and key findings in these sections.
References to Recent Literature: We have strengthened the introduction by citing the suggested articles and have expanded the bibliography with high-level references in English. We have kept only one Spanish reference, which is crucial for our study.
Specification of ICDAS: We used the ICDAS-II system and have clearly specified and described this in the methodology.
Null Hypothesis and Results: Both the null hypothesis and the primary and secondary outcomes have been incorporated into the introduction, and the discussion clearly states whether the null hypothesis was accepted or rejected.
Sample Size Calculation: The justification for the calculations has been described in greater detail in the methodology section, including reference values such as the mean and standard deviation.
Parental Knowledge on Oral Health: We have detailed the content of the questions, scoring process, question development, whether a pilot study was conducted, and how interviews and examinations were carried out in the methodology.
Figures and English Writing: We appreciate your comment and have corrected the figure legends and revised the entire manuscript to ensure it is properly written in English.
Discussion: We have expanded this section by thoroughly reviewing recent literature and placing our findings in a broader research context.
Manuscript Language Review: We apologize for the error and have corrected the sections of the manuscript that were in Spanish, ensuring proper use of English grammar and punctuation.
Thank you for your valuable comments, which have contributed to the improvement of our manuscript.
Reviewer 2 Report
Comments and Suggestions for Authors
- Line 122
It is stated: ,,Parental Oral Health Knowledge is Assessed through 14 multiple-choice questions’’ but in Results this is presented with 11 questions?
- ine 113 -114
,,Knowledge of oral health (14 binary-scored questions, categorized as either sufficient or insufficient knowledge as detailed below)’’ – Where is this?
- Line 124-126
,,Each correct answer was awarded 1 point. Scores were then categorized as Satisfactory: ≥70% correct answers (≥11 points) and Insufficient: <70% correct answers.’’ – Where is this categorization of answers presented in Results?
- Line 128-130
,,Parental Attitudes Toward Oral Health Assessed through a question related to the perceived importance of the child's oral health, categorizing parental attitudes as positive and negative or neutral.’’ - Parental Attitudes Toward Oral Health is presented with two questions in Results
- Line 132-133
,,Oral Health Practices at Home Evaluated through three questions about habits related to tooth brushing, tongue cleaning, and toothpaste use at home, categorized as good practices and poor practices.’’ – This is also not the case in Results.
- Line 136-138
,,Sociodemographic Variables. Additionally, the following complementary demographic variables were assessed: the child's gender, age of the parent/caregiver, educational level of the parent/caregiver, and the caregiver's relationship to the child.'' - Only Sociodemographic Variable presented in Results is child gender.
- Line 147
,,The researchers collected the data and received specific calibration for the ICDAS index provided by the principal investigator.'' – What is the level of agreement between examiners?
- Line 164, 165
It is stated that „non-parametric statistical tests were used: the Chi-square test (χ²), or Fisher's exact test“, but it is not evident in the Results, only „logistic regression“, which is not even mentioned in Data Management and Analysis. Also there is no odds ratio presented in Results as well.
- Line 188
There is no „the number of respodents - N“ in the Table 3.
- Line 193-200
Text is not written in English
- Figure 1.
Needs corrections and clarifications. If the Age group is formed with 9+ (like in Table 1.), why in Figure is presented group 10 years-old?
- Line 210.
In the text is mentioned that the analysis is done for fluoride use, and bacteria transmission as well, but there is no such data in the Table 4.
- The same questions in Table 4 are written in different ways
- Table 6.
It is stated that Table 6 „displays the relationship between reported oral hygiene practices, including brushing frequency and reaction to toothache“, but this is not so. Needs corrections and clarifications. Also, there is exactly the same question displayed in Table 4. and Table 6.
- Line 228-241
The text and the Figure 2 itself is very confusing. Needs corrections and clarifications or complete exclusion. Based on what criteria did the authors choose to present analysis of these three variables in Figure 2, why not other variables? No statistical significance is presented any ware in the Results? The Figure 2 includes Spanish words also.
- Line 243-248
Those Notes are sufficient
- Line 250-263
The text and the Figure 3 itself is also very confusing.
Further, it is stated that “the design clearly illustrates how protective behaviors (such as brushing three times a day, considering dental health essential, and recognizing the impact of sugar) are associated with a higher prevalence of children free from advanced caries“, but in the Figure 3 is presented only „sugar causes caries“ and „dental health is very important“. The Figure 3 is inconclusive.
- The Discussion is not specific. It is written in a rather general and vague manner. A significant result should be emphasized and discussed hierarchically before all other results, which is not the case in this manuscript. In addition, no statistically significant result is visible anywhere in the Results part of manuscript.
- Line 303-305
,,Regarding oral hygiene practices at home, our study identified daily tooth brushing as the protective factor most clearly associated with significantly reducing the risk of developing more advanced carious lesions (ICDAS 4–6). In particular, brushing two or more times a day showed a statistically significant association with lower rates of severe caries,, - Where in Results is this presented?
- The conclusions are not consistent with the Results, they are also unclear and imprecise. e.g. Where supervised brushing is analyzed in the Results?
- The References are not written in accordance with the Journal's guidelines and some references are repeated.
Author Response
We greatly appreciate the detailed comments from the reviewer, which have significantly enhanced the clarity and rigor of our manuscript. Below are our point-by-point responses:
Number of questions on oral health knowledge (Line 122): Thank you for pointing out this inconsistency. It was an unintentional error. After review, we have corrected this throughout the manuscript, and now both the methodology and results sections accurately indicate that 14 questions were used to assess oral health knowledge.
Binary scoring and knowledge categorization (Lines 113-114): We appreciate this observation. In this revised version, we have included an explicit explanation of the scoring and categorization system in the methods section. Additionally, we have added tables and illustrations to clarify the categorization.
Knowledge Categorization: satisfactory vs. insufficient: This has been corrected and the classification of scores as satisfactory (≥70%, ≥11 correct answers) or insufficient (<70%) is now detailed in Table 2 and clearly referenced in the Results section.
Parental Attitude Assessment: Thank you for identifying this discrepancy. You are correct; parental attitude towards oral health was assessed with a single question. This clarification is made explicit in the revised version to avoid confusion.
Oral health practices at home: The evaluation was based on two questions: "Child's feeling when brushing" and "Child brushes their tongue." We have corrected the Methods and Results sections to ensure clarity and coherence.
Sociodemographic variables: The sociodemographic data are described in the Results and presented in Table 1, emphasizing gender and age as the most relevant variables for this analysis.
Examiner calibration for the ICDAS index: The details of the calibration and inter-examiner agreement for the ICDAS index are now described in the Methods section of this version, as recommended for studies of this type.
Non-parametric statistical tests and logistic regression: In the Methods section, we provide a detailed description of the statistical analyses performed, including non-parametric tests (chi-square, Fisher's exact test) and logistic regression. The odds ratios are properly presented in the results as recommended.
Mismatch between age groups (Figure vs. Table 1): The correct age group is up to 9 years, and we have reviewed the figure to ensure consistency between figures and tables.
Fluoride use and bacterial transmission analysis: We clarified that analyzing the effectiveness of fluoride use was not an objective of this study. The tables and results have been revised to reflect only the investigated variables.
Table 6 – Relationship between brushing frequency and toothache: The tables have been revised to avoid the repetition of questions, and Table 6 now presents relationships specifically related to brushing frequency and toothache, with textual references updated accordingly.
Figure 2 – Clarity and text in Spanish: Figure 2 has been revised for greater clarity, inclusion criteria for selected variables have been added, all Spanish text has been removed, and the presentation of statistical significance has been improved.
Figure 3 – Conclusions on protective behaviors: We have created new tables and figures for better presentation of the article and to ensure that all relevant behaviors are adequately represented.
Thank you again for your valuable comments, which have significantly improved the quality and clarity of our manuscript.
Reviewer 3 Report
Comments and Suggestions for Authors
Review
Manuscript title: ‘How Parental Knowledge, Attitudes, and Practices Shape Childhood Dental Caries: Evidence from a Rural Ecuadorian Population Using the ICDAS Index’
Abstract
• ‘Results revealed a high prevalence of advanced caries (73.4%) and a statistically significant association between frequent tooth brushing and a reduced risk of cavitated lesions.’
Provide numerical data related to the second statement ( ’statistically significant association…’)
Are these results related to the regression analysis? Please, indicate.
Introduction
• Line 41-43 ‘During the school years, children develop autonomy, language, and habits, making oral care a priority for professionals.’
Make the statement clearer.
• Line 50-51 ‘Parental education, socioeconomic level, and prevention awareness are associated with higher caries rates (26,27).’
The direction of the association in this statement is not clear. The reader could assume, for example, that the better the parental awareness, the higher the caries rates… Improve the statement (e.g., remove ‘higher’).
Methods
• Line 77: ‘An observational, cross-sectional, and prospective study was conducted.’
The study design looks to me as a cross-sectional study. This is not a prospective study!
• Line 93-94 ‘ Children with severe systemic diseases’… ‘Children receiving ongoing specialized dental treatments’
Provide clarification to both statements (which severe diseases? What does it mean by ‘specialized dental treatments?)
• Line 122-123 ‘Parental Oral Health Knowledge Assessed through 14 multiple-choice questions about oral hygiene, fluoride use, dietary practices, dental visits, and transmission of cariogenic microorganisms.’
What was the question related to the ‘transmission of cariogenic microorganisms’, and which answer related to this question was assumed as the correct answer?
• Line 147-148 ‘The researchers collected the data and received specific calibration for the ICDAS index provided by the principal investigator.’
Clarify how many examiners were involved. Report the agreement data after calibration if available.
Results
• Line 179 ‘17.0% (39 children) showed initial caries (ICDAS 1-3),’
The results look strange since usually the prevalence of non-cavitated lesions is always higher than cavitated lesions when the contemporary caries diagnosis criteria are used.
• Questionnaire ‘Amount of toothpaste used’
Why is there no option of a pea size? The rice size is recommended till 3 years of age, isn’t it?
• Line 193-199 and Fig. 1: not translated to English!
• I have concerns about the appropriateness of the logistic regression analysis. I do not see much sense in calculating the regression coefficient for each association of interest without adjusting for the confounding factors/covariates.
• The results of Figure 3 are unclear to me.
• I did not find the results reported when Chi-square and Fisher's exact test were used
• Overall, the paper needs major revisions.
Author Response
We greatly appreciate the detailed comments from the reviewer, which have significantly improved the clarity and rigor of our manuscript:
-
Lines 41-43: "During the school stage, children develop autonomy, language, and habits, making oral care a priority for professionals." Please clarify this statement: Thank you for your suggestion. We have rewritten this section to provide a clearer statement. It now explicitly highlights the importance of oral health care in child development and the key role health professionals play in promoting these habits. The revised sentence says: "During the school years, children gain greater autonomy and develop language and self-care habits. In this context, promoting proper oral health habits should be considered a priority for health professionals."
-
Lines 50-51: "Parental education, socioeconomic level, and awareness of prevention are associated with higher rates of caries.” Thank you for this observation. We recognize that the previous wording could cause confusion regarding the direction of the association. We have corrected it to clarify that lower parental education, lower socioeconomic level, and lower awareness of prevention are associated with higher rates of caries.
-
Line 77: "An observational, cross-sectional, and prospective study was conducted." The study design seems to be cross-sectional, not prospective: We appreciate your precision. It was a wording error. The description has been corrected to indicate that the study is strictly observational and cross-sectional.
-
Lines 93-94: "Children with severe systemic diseases" ... "Children undergoing ongoing specialized dental treatments." Please clarify both statements: Thank you for pointing out the need for clarification. We have now specified that we excluded children with severe systemic diseases such as uncontrolled diabetes mellitus, congenital heart disease, or immunodeficiencies, as well as those undergoing specialized dental treatments such as active orthodontic therapy or rehabilitation after major dental trauma, to avoid biases in the sample.
-
Lines 122-123 Comment: "Parental knowledge about oral health: evaluation with 14 multiple-choice questions on oral hygiene, fluoride use, eating habits, dental visits, and transmission of cariogenic microorganisms." What was the question related to the “transmission of cariogenic microorganisms,” and what was the correct answer?: Thank you for your comment. This was actually a wording error in the manuscript. The questionnaire assessed knowledge, attitudes, and practices; the question about microorganism transmission referred to the awareness of the possibility of transmitting cariogenic bacteria from parents to children through habits like sharing utensils, with the correct answer acknowledging this possibility. This section has been revised to accurately reflect the content of the questionnaire.
-
Lines 147-148: "Researchers collected the data and received specific ICDAS index calibration from the principal investigator.": Thank you for noting this. We have clarified in the methodology that the researchers underwent a specific and structured calibration process for using the ICDAS index, including details about the duration and procedures of the training.
-
Line 179: "17.0% (39 children) had initial caries (ICDAS 1-3)..." The results seem strange, as typically, the prevalence of non-cavitated lesions is higher than that of cavitated lesions when using contemporary caries diagnosis criteria: Thank you for your comment. This point has been clarified in the manuscript, indicating that the study population comes from a rural, agricultural, and low-income area of Ecuador, which may influence the presence of more advanced (cavitated) lesions due to limited access to preventive care and late identification of initial lesions.
-
Questionnaire: "Amount of toothpaste used" Comment: Why can't the "pea-sized" option be selected? The recommendation is rice-sized up to 3 years, isn't it?: Thank you for your observation. You are correct, and we appreciate your insight. We used a prevalidated questionnaire, so we initially did not modify the options.
-
Logistic regression analysis and statistical results: adequacy of the logistic regression analysis: Thank you for your valuable observations and suggestions. We have made significant revisions to both the methodology and the results. The statistical analysis is now presented more appropriately, including adjustments for potential confounders and a detailed presentation of the chi-square and Fisher's test results. Moreover, Figure 3 has been revised to ensure greater clarity and relevance.
We thank you for your timely observations, which have led to significant improvements in the quality of the manuscript.
Round 2
Reviewer 1 Report
Comments and Suggestions for Authors
I read the revised manuscript, formerly titled "How Parental Knowledge, Attitudes, and Practices Shape Childhood Dental Caries: Evidence from a Rural Ecuadorian Population Using the ICDAS Index," and now titled "Parental Knowledge, Attitudes, and Practices and Their Association with Dental Caries in Children Aged 5–9 Years: A Cross-Sectional Study in Rural Ecuador," with great interest.
The authors have made sufficient revisions in line with my previous comments, particularly concerning the title, abstract, and methodological aspects, demonstrating significant effort.
Considering the wide age range, the importance of parental knowledge in the context of dental caries, and the fact that dental caries remains a major global public health issue, I believe this manuscript makes a valuable contribution to the literature. Furthermore, given the limited number of studies on this topic conducted in Ecuador, a relatively small country, the manuscript also represents a noteworthy contribution to the local body of research.
Taking into account all the revisions made by the authors, I am of the opinion that the manuscript is suitable for publication in the journal. The revisions are sufficient. I congratulate the authors on their successful research and the resulting manuscript.
Author Response
Dear Editor,
We sincerely appreciate your positive comments on our manuscript. We are pleased to know that you found the revisions to the title, abstract, and methodological aspects significant. Your acknowledgment of the manuscript's contribution to the scientific literature and the local body of research in Ecuador is highly valued by our team.
We are thrilled that you consider our work suitable for publication and are grateful for your support and guidance throughout the review process. If there are any further comments or suggestions, we are ready to continue improving.
Thank you once again for your support and recognition.
Sincerely,
María Saquicela-Pulla
Corresponding Author.
Reviewer 2 Report
Comments and Suggestions for Authors
The study design, materials and methods, results and discussion have been significantly
improved and now are more accurate.
Minor corrections to the manuscript are required.
Line 231
Sex and gender are not the same, correct this.
Line 240 – 242, 368…
Uniformize ICDAS coding throughout the manuscript.
Line 244
Correct Typo (“n”)
Line 254, 257 and Table 3
The same question like in the first review: “Where is the number of respondents (N)?”
Line 272-275.
The controversial result about the response “Between meals” as the best time to consume sweets indicating a significant negative association with the presence of caries—interpreted as a potentially protective behavior should be discussed!
Line 367
This is confusing, is it child’s or caregivers' oral hygiene practices?
Line 408
The educational level of the caregivers is discussed, but where in manuscript are these results presented?
Line 423
What is CPOD? Explain.
Author Response
Dear Editor,
We sincerely appreciate the valuable comments and observations made by the reviewers regarding our manuscript. We have carefully reviewed each suggestion and made the corresponding corrections in the text, detailed point by point below. We believe that these improvements have significantly strengthened the quality and clarity of the manuscript.
Responses to editorial comments:
-
Line 231 – Differentiation between sex and gender The text was corrected to properly differentiate between sex and gender, using the term "sex" in accordance with current recommendations.
-
Lines 240–242, 368 – Standardization of ICDAS coding The use of the categories "ICDAS 0–3" and "ICDAS 4–6" was standardized throughout the manuscript, eliminating typographical inconsistencies.
-
Line 244 – Typographical correction of “n” The error was corrected and replaced with an uppercase "N," according to the standard for representing the total number of participants.
-
Lines 254, 257, and Table 3 – Inclusion of the number of respondents (N) A footnote was added to the table indicating the total number of caregivers surveyed (N = 229), with a specification of percentage calculation per item. Corrections were also made in Table 3, which previously showed only percentages.
-
Lines 272–275 – Discussion on the response “between meals” A paragraph was added in the discussion section explaining that the negative association of this response with caries presence might be due to ambiguous interpretation or complex dietary patterns, recommending caution in interpretation.
-
Line 367 – Clarification about hygiene practices It was specified that the hygiene practices evaluated pertain to the children, improving the wording to avoid ambiguity.
-
Line 408 – Inclusion of results on educational level In the results section, statistical analysis was added, indicating no significant association between the caregiver's educational level and caries prevalence.
-
Line 423 – Definition of CPOD An explanation was added at the first use of the term that the CPOD index represents "Decayed, Missing, and Filled Teeth," differentiating it from the ICDAS-II system.
All corrections have been highlighted in the revised document, as requested. Additionally, scientific coherence, journal style, and the integrity of the original manuscript have been maintained.
We remain attentive to any further observations and thank you in advance for reconsidering the manuscript for publication.
Sincerely, María Saquicela-Pulla
Corresponding Author